# Modelling the acclimation capacity of coral reefs to a warming ocean

**Nomenjanahary Alexia Raharinirina**[1,2¤a], **Esteban Acevedo-Trejos**[1¤b], **Agostino Merico**[1,2]*

**1** Department of Integrated Modelling, Leibniz Centre for Tropical Marine Research, Bremen, Germany,
**2** Department of Physics & Earth Sciences, Jacobs University Bremen, Bremen, Germany

¤a Current address: Mathematics of Complex Systems, Zuse Institute Berlin, Berlin, Germany
¤b Current address: Earth Surface Process Modelling, German Research Centre for Geosciences, Potsdam, Germany
* agostino.merico@leibniz-zmt.de

**Data Availability Statement:** The model is coded in Python and is provided as open source software under the GNU General Public License, version 2. It can be downloaded from GitHub (https://github.com/systemsecologygroup/CoralZooxProjection)

## Abstract

The symbiotic relationship between corals and photosynthetic algae is the foundation of coral reef ecosystems. This relationship breaks down, leading to coral death, when sea temperature exceeds the thermal tolerance of the coral-algae complex. While acclimation via phenotypic plasticity at the organismal level is an important mechanism for corals to cope with global warming, community-based shifts in response to acclimating capacities may give valuable indications about the future of corals at a regional scale. Reliable regional-scale predictions, however, are hampered by uncertainties on the speed with which coral communities will be able to acclimate. Here we present a trait-based, acclimation dynamics model, which we use in combination with observational data, to provide a first, crude estimate of the speed of coral acclimation at the community level and to investigate the effects of different global warming scenarios on three iconic reef ecosystems of the tropics: Great Barrier Reef, South East Asia, and Caribbean. The model predicts that coral acclimation may confer some level of protection by delaying the decline of some reefs such as the Great Barrier Reef. However, the current rates of acclimation will not be sufficient to rescue corals from global warming. Based on our estimates of coral acclimation capacities, the model results suggest substantial declines in coral abundances in all three regions, ranging from 12% to 55%, depending on the region and on the climate change scenario considered. Our results highlight the importance and urgency of precise assessments and quantitative estimates, for example through laboratory experiments, of the natural acclimation capacity of corals and of the speed with which corals may be able to acclimate to global warming.

## Author summary

Tropical coral reefs are among the most productive and diverse ecosystems on Earth. The success of these ecosystems depends on a symbiotic relationship between corals and unicellular algae. This relationship breaks down when water temperature increases above certain levels causing massive coral deaths. Therefore, the future of coral reef ecosystems

along with the literature data used to parametrise bleaching (Table 2 and S1 Appendix. Levels of bleaching). The coral cover data used to estimate the speed of coral acclimation (presented in S2 Appendix. Speed of acclimation) were processed from raw datasets and are all available in the GitHub repository mentioned above (code file: "N_estimation.py"). The light green dots are average values of the raw coral cover data published in Bruno and Selig (2007) and provided to us by John Bruno via personal communication. The dark green dots are median values of raw cover data published in De'ath et al. 2012.

**Funding:** N.A.R. was funded by the Leibniz Centre for Tropical Marine Research (ZMT), via institutional funding, and by the Deutscher Akademischer Austauschdienst (DAAD), via STIBET III. A.M. was funded by the Leibniz Centre for Tropical Marine Research, via institutional funding. E.A.-T. was funded by the Deutsche Forschungsgemeinschaft (DFG) through the grant AC 331/1-1. The funders had no role in study design, data collection and analysis, decision to publish, or preparation of the manuscript.

**Competing interests:** The authors have declared that no competing interests exist.

depends on the capacity of corals to acclimate to current warming rates. Despite many studies have tried to predict the future of coral reefs, these predictions are impaired by uncertainties related to the speed with which corals can acclimate. We developed a model in which corals can acclimate to changing temperature. By comparing model results with observations of coral cover, we estimated the speed of coral acclimation at the community level in different regions of the tropics. Using this information, we quantified the future changes in coral abundances under different warming scenarios. We found that corals of the Great Barrier Reef have higher acclimation capacities than those of other regions. Our results showed substantial coral declines in South East Asia and Caribbean, especially under the highest warming scenarios. Thus, we provide evidence that natural acclimation alone may not be sufficient to offset the decline of corals caused by the expected warming trends.

## Introduction

Shallow water coral reefs are marine limestone structures accreted by tiny organisms called polyps. They are characteristic of the tropical oceans, form habitats for a myriad of other organisms [1–3], and provide important ecological services, including food and coastal protection, for millions of people [4–6]. Coral polyps (corals henceforth) form a symbiotic relationship with unicellular photoautotrophs (the symbiont), called zooxanthellae [7, 8]. This association provides corals (the host) with photosynthetically fixed carbon [9, 10] and has thus contributed to their success in the nutrient-poor waters of the tropics [8, 11, 12]. Elevated sea temperature causes a breakdown of the coral-algae symbiosis [13–17]. In the worse cases, this breakdown leads to the expulsion of algae by the coral host, a process known as bleaching [13, 18, 19] because the loss of photosynthetic pigments makes the white skeleton of corals visible through the transparent tissue. Under the current rates of global warming [20, 21], a critically important aspect for the future of coral ecosystems is to determine whether corals will be able to respond at a sufficiently fast pace.

Corals can respond to global warming in different ways, including by moving to more favourable habitats, by genetic adaptation, or by acclimation. For sessile species like corals, range shifts are constrained by dispersal during their larval stage, the movements of which are strongly influenced by the prevailing currents [22]. Genetic adaptation is characterised by allele frequency changes between generations and is mediated by natural selection [23]. In contrast, acclimation is a response that does not involve genetic changes. It is characterised by phenotypic plasticity, whereby an alteration of the organism's phenotype occurs in response to environmental change within the lifetime of the individual [23, 24]. Examples of phenotypic plasticity in corals include altered heterotrophic feeding during bleaching [25], increased retention of symbionts' chlorophyll during heat-stress [26], enhanced photosynthetic rate [27], and induction of heat shock proteins [28] during light-stress, heat-stress, or both.

Previous modelling studies addressed coral adaptation either in a diagnostic fashion, as a threshold response mechanism [29–31], or by considering genetic responses of symbionts, encompassing an increase in thermal tolerance [32]. The thermal acclimation or adaptation capacity of coral reefs has been considered previously in two modelling studies [30, 31] and was based on empirical thermal bleaching thresholds, between 1˚C-months and 6˚C-months, derived from the most recent mean of maximal monthly temperatures, because these temperatures have shown damaging effects in coral reefs. However, the mechanisms underpinning thermal acclimation or adaptation, which depend on the physiological machinery of corals, are

not considered in these previous modelling studies. In addition, little is known about the speed of coral acclimation, an aspect that is critical for producing reliable predictions about the fate of coral reefs in a changing climate.

Here we present a new model that captures a whole coral community by considering the average energy invested in the symbiotic relationship as a mean community trait fuelled by the physiological machinery of the corals. We then use this model in combination with observations of changes in coral cover over time to estimate the speed of coral acclimation at the community level. Finally, we investigate the effects that different global warming scenarios have on three coral reef systems of the tropics: the Great Barrier Reef, South East Asia, and the Caribbean (Fig 1). In our model (Fig 1a), coral growth is constrained by temperature-limited growth curves (Fig 1b) and corals respond to changing environmental temperature via acclimation (i.e. phenotypic plasticity). We make the distinction between "thermal acclimation and/or adaptation", which is defined as an increase in bleaching thresholds [30, 31], and physiological acclimation, which we define as an increase in coral fitness under changing temperature. Physiological acclimation in nature encompasses phenotypic changes that individual corals can undergo in order to compensate for the growth deficiencies induced by thermal stress, these changes may include increased retention of chlorophyll [26], induction of heat shock proteins [28], and other physiological and immunological responses [33]. Our model does not resolve single species or individual organisms but it captures the effects that such changes may have at the community level through variations in the average trait.

The novelty of our work lays in the way we implement coral acclimation to changing temperature. In our model, community variations in physiological acclimation are captured by changes in an average physiological trait, which is defined as the mean energy that the coral community invests in the symbiotic relationship, and is proportional to the gradient of coral fitness. The constant of proportionality denotes the speed of coral acclimation, i.e. the speed with which the coral community moves towards an optimal trait value, one that maximises fitness, under changing environmental temperature. Future projections for bleaching, energy investment trait, and coral and symbiont abundances are produced under three Representative Concentration Pathways (RCPs) of future $CO_2$ emissions: RCP 2.6, RCP 4.5 and RCP 8.5, respectively for low, moderate and high $CO_2$ emissions.

To foster reproducibility, transparency, and flow of ideas, we provide the numerical code of our model as open-source software (https://github.com/systemsecologygroup/CoralZooxProjection) so that it can be used, modified, and redistributed freely.

## Models and methods

We developed a trait-based mathematical model to investigate the acclimation dynamics of coral-algae symbiosis under different Representative Concentration Pathways (RCPs) and in different regions of the tropics. The model is based on evidence that corals are in control of the symbiotic relationship [37–39]. The model describes the temporal dynamics of the coral abundance ($C$), of the symbiotic algal abundance ($S$), and of the coral trait ($U$).

Since our study is focused on coral acclimation as a mechanism for counteracting global warming [40, 41], we choose to consider a coral trait ($U$) that is subject to plastic change over ecological time scales. This plastic trait is defined as the mean energy per coral abundance that a coral community invests per unit of time in the symbiotic relationship (for example, the energy necessary for providing $CO_2$ to algae [34–36]).

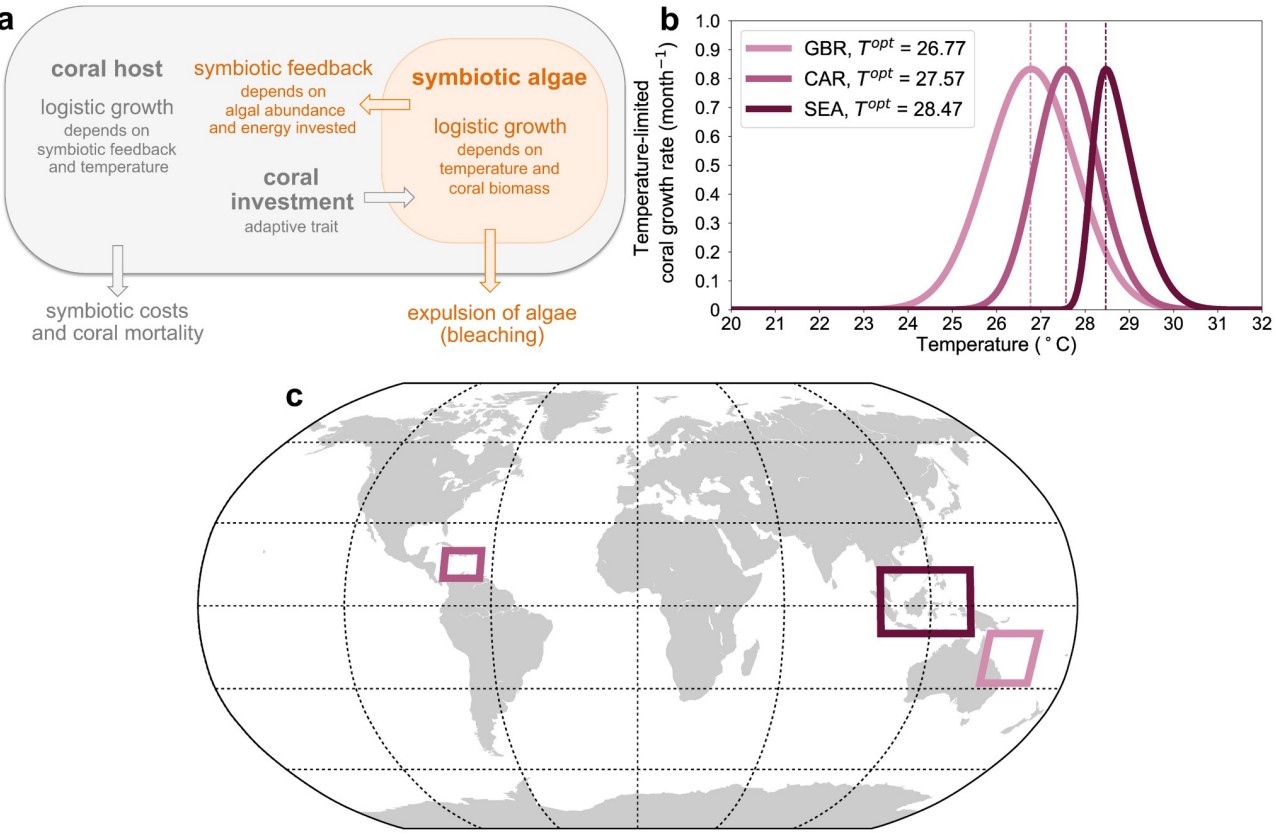

**Fig 1. Model schematic and studied regions.** (a) Schematic depicting the major interactions between the coral host and the symbiotic algae captured by the model; the coral host grows logistically as a function of symbiotic feedback and environmental temperature; corals invest energy in symbiosis (for example the energy necessary for providing $CO_2$ to the algae [34–36]) and receives a symbiotic feedback (for example photosynthate); corals sustain symbiotic costs, which are associated to the maintenance of microenvironments for housing the algae [34] and to stress associated to photosynthesis (like the presence of reactive oxygen species [35]), and are subject to mortality; algae grow logistically as a function of environmental temperature and coral abundance; algae are expelled during bleaching events, which occur when environmental temperature exceeds the optimal regional temperature (i.e., $T^{opt}$, panel **b**) for coral growth by a variable range of temperature increase reflecting the severity of bleaching (see subsection Bleaching). (b) Temperature-limited coral growth curves (Eq 7); dashed lines mark the regional temperature optima. (c) Studied regions: Great Barrier Reef (GBR), South East Asia (SEA), and Caribbean (CAR). The map is produced with the Python's Matplotlib Basemap Toolkit available from https://matplotlib. org/basemap/api/basemap_api.html (© 2011, Jeffrey Whitaker).

The model is formulated as follows:

$$\text{Coral} \qquad \frac{dC(t)}{dt} = F(U,\, S,\, C) \cdot C(t), \qquad (1)$$

$$\text{Symbiont} \qquad \frac{dS(t)}{dt} = G_s(T) \cdot \left(1 - \frac{S(t)}{K_S(C)}\right) \cdot S(t), \qquad (2)$$

$$\text{Trait} \qquad \frac{dU(t)}{dt} = N \cdot \frac{\partial F(U,\, S,\, C)}{\partial U}, \qquad (3)$$

where

$$\text{Coral fitness} \qquad F(U,\, S,\, C) = G_C(T) \cdot \kappa(S) \cdot E(U) \cdot \left(1 - \frac{C(t)}{K_C}\right) - \mu(U,\, S) - M_C, \qquad (4)$$

$$\text{Symbiotic feedback} \qquad \kappa(S) \cdot E(U) = \left( \frac{S(t)}{\Gamma_h \cdot C(t) + S(t)} \right) \cdot \left( 1 - e^{-\beta \cdot U(t)} \right), \qquad (5)$$

$$\text{Symbiotic cost} \qquad \mu(U, S) = \alpha \cdot e^{r \cdot U(t)} \cdot \frac{S(t)}{K_S(C)}. \qquad (6)$$

The full set of model variables, parameters, units, and values are reported in Table 1. The different components and specific terms of the model are detailed in the following.

## Coral dynamics

In our model, net coral growth (Eq 4) is simulated as the difference between gross growth, $G_C \, \kappa E (1 - C/K_C)$, and losses due to symbiotic costs (Eq 6) and natural mortality, which is

**Table 1. State variables, functions, and parameters constituting our model.**

| Symbol | Description | Units | Value | Reference |
|---|---|---|---|---|
| $C$ | coral abundance | cm$^2$ | variable | |
| $S$ | algal abundance | cells | variable | |
| $U$ | coral plastic trait (energy investment) | energy $\cdot$ cm$^{-2}$ $\cdot$ month$^{-1}$ | variable | |
| $t$ | time | month | variable | |
| $F$ | fitness | month$^{-1}$ | function | |
| $\kappa$ | fraction of coral growth dependent on $S$ | dimensionless | function | |
| $E$ | fraction of coral growth dependent on $U$ | dimensionless | function | |
| $\mu$ | cost of symbiosis | month$^{-1}$ | function | |
| $G_C$ | temperature-limited coral growth rate | month$^{-1}$ | function | |
| $G_S$ | temperature-limited symbiont growth rate | month$^{-1}$ | function | |
| $G_{max}$ | coral maximum growth rate | month$^{-1}$ | 0.83 | [42] |
| $a$ | symbiont linear growth parameter | month$^{-1}$ | 0.09 | [10, 32] |
| $b$ | symbiont exponential growth parameter | °C$^{-1}$ | 0.063 | [43, 44] |
| $K_C$ | coral carrying capacity | cm$^2$ | $4.4 \cdot 10^{15\dagger}$, $1.2 \cdot 10^{16\ddagger}$, $1.9 \cdot 10^{15\S}$ | |
| $K_S$ | symbiont carrying capacity | cells | function | |
| $K_{smax}$ | maximum symbiont carrying capacity per coral abundance | cells $\cdot$ cm$^{-2}$ | $3 \cdot 10^6$ | [45] |
| $M_C$ | coral mortality rate | month$^{-1}$ | $0.83 \cdot 10^{-3}$ * | |
| $G$ | coral growth function | dimensionless | function | |
| $T$ | environmental temperature (forcing) | °C | external input | |
| $\bar{T}$ | mean of growth temperature $G$ | °C | $26.8^\dagger$, $28.1^\ddagger$, $27.1^\S$ | [46] |
| $T^{opt}$ | optimal temperature for coral growth | °C | $26.8^\dagger$, $28.5^\ddagger$, $27.6^\S$ | |
| $\sigma$ | standard deviation of $G$ | °C | $1.0^\dagger$, $0.8^\ddagger$, $0.9^\S$ | [46] |
| $s$ | skewness of $G$ | °C | $2 \cdot 10^{-4\dagger}$, $3.8^\ddagger$, $1.1^\S$ | [46] |
| $\alpha$ | coral linear cost parameter | month$^{-1}$ | $10^{-3}$ * | |
| $r$ | coral exponential cost parameter | (energy $\cdot$ cm$^{-2}$ $\cdot$ month$^{-1}$)$^{-1}$ | $12 \cdot 10^3$ * | |
| $\beta$ | strength of symbiotic feedback | (energy $\cdot$ cm$^{-2}$ $\cdot$ month$^{-1}$)$^{-1}$ | $12 \cdot 10^2$ * | |
| $\Gamma_h$ | symbiont to host ratio for which $\kappa = 0.5$ | cells cm$^{-2}$ | $1 \cdot 10^{6\|}$ | [45] |
| $N$ | speed of acclimation (estimated in this study) | (energy $\cdot$ cm$^{-2}$ $\cdot$ month$^{-1}$)$^2$ | $5.54 \cdot 10^{-13\dagger}$, $2.65 \cdot 10^{-13\ddagger}$, $2.375 \cdot 10^{-13\S}$ | |

* Parameter not quantified in the literature, its value was thus chosen to obtain a good model to data fit and to produce stable dynamics.

† Value for Great Barrier Reef;

‡ Value for South East Asia;

§ Value for the Caribbean.

‖ Minimal value for healthy corals.

captured by the parameter $M_C$, which accounts for losses other than those related to the maintenance of the symbiotic relationship (e.g., due to senescence or physical damages).

Gross growth, $G_C \kappa E (1 - C/K_C)$, is the product between a temperature-dependent growth rate $G_C$, a symbiotic feedback $\kappa E$ (Eq 5), and a logistic term. The logistic term, which is defined by the carrying capacity $K_C$, is the most commonly used formulation for limited growth. We calculated the carrying capacities of a specific region by summing up the planar surface area of all the potential reef habitats estimated by a previous study [47] in that region, and we accounted for the shape of corals by multiplying these values with the mean of the conversion factors, 11.86 and 16.4 for, respectively, massive and branching corals [48]. This approach for calculating coral carrying capacities allows us to account for the contribution, in roughly equal proportions, of the two most common coral morphologies. In reality, these morphologies may not always occur in equal proportions. Albeit with this caveat, a previous study [48] determined that this approach produces more realistic estimates than simply considering planar habitat areas.

The symbiotic feedback $\kappa E$ (Eq 5) reflects the benefit that corals receive from algae. $\kappa$ is the fraction of coral growth due to the translocation of photosynthate and thus dependent on symbiont abundance. $\kappa$ tends to a maximum 1 as the symbiont abundance $S$ increases, and half saturates as the symbiont abundance reaches a certain fraction $\Gamma_h$ of coral abundance $C$. $E$ is the fraction of coral growth dependent on the amount of energy $U$ invested in symbiosis. $E$ increases exponentially with $U$ at a rate $\beta$ and saturates to 1, reflecting the fact that benefits received by corals cannot increase indefinitely. It follows that the symbiotic feedback $\kappa E$ is 0 in the absence of symbionts and/or in the absence of energy investment $U$ and is maximised when the symbiont abundance $S$ and the energy investment $U$ reach optimal values.

At minimum symbiotic feedback (i.e. when $\kappa E = 0$), corals grow at rate 0; at maximum symbiotic feedback (i.e. when $\kappa E = 1$), corals grow at rate $G_C$, which depends on environmental temperature $T$, as follows:

$$G_C(T) = G_{max} \cdot \left( \frac{G(T)}{\max[G(T)]} \right), \tag{7}$$

where $G(T)/\max[G(T)]$ is the normalised coral growth function and $G_{max}$ is the maximum possible coral growth rate. Fig 1b shows the temperature-limited growth rate curves $G_C(T)$ of the different regions. If environmental temperature falls away from these tolerance ranges then coral growth will be zero and the abundance of the community will decline.

Given that the extension rates of corals range between 0.08 and 20 cm year$^{-1}$, depending on the coral species and their location depth [42], we choose a coral maximum growth rate $G_{max}$ equal to 10 year$^{-1}$, i.e. 0.83 month$^{-1}$, to reflect the maximum growth rate (in terms of fold change per unit of time) of coral communities composed of both massive and branching corals. Thus, $G_C$, represents the proportion of maximal growth that the coral can achieve given the environmental temperature and given the symbiotic association with the algae.

The global occurrence of coral reefs as a function of temperature, can be represented by a skewed normal distribution, with mean $\bar{T}$, standard deviation $\sigma$, and skewness $s$ [46]. Thus we assumed

$$G(T) = \phi \cdot \left( \frac{T - \bar{T}}{\sigma} \right) \cdot \Phi \cdot \left( s \cdot \frac{T - \bar{T}}{\sigma} \right), \tag{8}$$

where $\phi$ and $\Phi$ are, respectively, probability and cumulative distribution functions of a normal distribution with mean 0 and standard deviation 1. This formulation associates temperature-limited coral growth rates to fixed thermal tolerance distributions.

The costs incurred by the coral host for investing energy into the symbiotic relationship are represented by the term $\mu(U, S)$ (Eq 6). We distinguish between costs associated merely to the presence of the symbiont (thus depending on $S$) and costs related to the energy invested in the symbiotic relationship (thus depending on $U$). The costs associated to the presence of the symbiont (e.g. costs associated to the production of peri-algal vacuoles [8] are represented by a linear term ($S/K_S$). This formulation simulates the presence of a physiological limit on the size of the symbiont population that corals can sustain. The costs related to the energy invested in the symbiotic relationship, e.g. damages from reactive oxygen species [35], are non-saturating and increase exponentially at a rate $r$. $\alpha$ is a proportionality term bearing the unit of the symbiotic cost. It follows that when $U = 0$, then $\mu \neq 0$ if $S \neq 0$, reflecting the fact that corals still bear costs related to the mere presence of symbionts even when no energy is invested in the symbiotic relationship. These can be considered as baseline operating costs for corals. When $S = 0$, then $\mu = 0$, reflecting the lack of symbiotic costs in the absence of symbionts. In this case, corals do not receive any symbiotic benefit and die at a mortality rate $M_C$.

## Symbiont dynamics

Since the growth of zooxanthellae is controlled by the coral host [37–39], we assumed that the symbiont population grows logistically (Eq 2) with a maximum growth rate $G_S$ and a carrying capacity $K_S$ that depends on coral abundance $C$. In analogy to other unicellular photoautotrophs, the maximum growth rate of the symbiont depends on temperature according to the following function:

$$G_S(T) = a \cdot e^{b \cdot T}, \tag{9}$$

where $T$ is environmental temperature, $a = 1.0768$ year$^{-1}$, i.e. 0.09 month$^{-1}$ [10, 32], and $b = 0.063°C$ [43]. Eq 9 is an envelop function for temperature-dependent growth rates expressed by multiple phytoplankton species in different laboratory cultures [43].

The carrying capacity is defined as:

$$K_S = K_{smax} \cdot C, \tag{10}$$

where $K_{smax}$ is the amount of symbiont abundance per coral abundance found, on average, in healthy coral communities. $K_{smax}$ usually ranges between one and six millions cells per cm$^2$ of coral surface, depending on the coral species [45, 49]. We, therefore, set $K_{smax} = 3 \cdot 10^6$ cells · cm$^{-2}$.

The full control of algal growth by the corals is attained by preventing zooxanthellae to take up any photosynthate they produce and by assuming an infallible ability of the host to control the nutrient flux. The latter ensures that the symbiont population never exceeds the hosting capacity of the corals. In nature, eutrophication can concur with temperature to induce a breakdown of the symbiotic relationship through uncontrolled symbiont growth [50]. These synergistic effects, however, are beyond the scope of our study.

## Bleaching

Bleaching is defined as the loss of zooxanthellae cells and/or zooxanthellae pigments [51, 52]. This phenomenon is caused by a variety of factors, the most important being increases in sea temperature and solar radiation [19, 53]. Evidence indicates that the loss of cells is driven by high temperature, whereas the loss of pigments is driven by high light [54]. Since our work mainly focuses on bleaching induced by thermal stress, we simulate this process as the loss of zooxanthellae cells.

Temperature thresholds for bleaching are typically estimated as the level of thermal stress, which are known as Degree Heating Weeks (DHW) [15, 55, 56] (see also https://coralreefwatch.noaa.gov/satellite/index.php) or Degree Heating Months (DHM) [30]. These metrics represent different ways of measuring accumulated SSTs above a climatological maximum [30, 55]. They account for both magnitude and duration of thermal stress, which are the determining factors of the severity of bleaching events [8]. These metrics do not consider the amount of zooxanthellae abundance lost, but rather indicate the percentage of corals at risk of degradation due to bleaching [30, 55]. In addition, these metrics account only for the effect of temperature despite solar irradiance is also a decisive factor. New methods, that describe bleaching by the combined effect of solar irradiance and temperature, are being developed [57]. Here, we use a much simpler method. We implement bleaching as a reduction $\delta_S$ in symbiont abundance when, at a given time $t$, the environmental temperature $T(t)$ exceeds the optimal temperature for coral growth $T^{opt}$ by some amount $\epsilon_T$ falling within the observed ranges of temperature increase $\Delta T_{obs}$ (Table 2). Due to the high variability shown by the observations (S1 Appendix), the percent reduction of symbiont abundance $\delta_S$ corresponding to $\Delta T_{obs}$ is randomly drawn from a uniform distribution $\mathcal{U}$ constrained by the observed ranges of symbiont reduction $\Delta S_{obs}$ (Table 2). In summary, a bleaching event at a time $t$ is included in the symbiont dynamics by imposing the following condition on the symbiont abundance:

$$S(t) = \begin{cases} S(t) & \text{if} \quad T(t) < T^{opt} + \epsilon_T\,; \quad \text{with } \epsilon_T \in \Delta T_{obs} \\ \delta_S \cdot S(t) & \text{if} \quad T(t) \geq T^{opt} + \epsilon_T\,; \quad \text{with } \epsilon_T \in \Delta T_{obs} \text{ and } \delta_S \sim \mathcal{U}(\Delta S_{obs}) \end{cases} \quad (11)$$

Being based on observations of symbiont abundance reduction following bleaching events, our method reflects the long-term evolutionary adaptation to local temperature.

## Coral acclimation

In contrast to a previous work [58] that simulated the genetic adaptation of corals, a novel and crucial feature of our model is the focus on coral acclimation to changing temperature. In our model, corals can acclimate within fixed temperature-limited growth curves (Fig 1b), which are expressions of long-term evolutionary adaptations to local temperature variations. An approach based on genetic adaptation would involve, instead, shifts in thermal growth optima $T^{opt}$, i.e. changes in the temperature-limited coral growth curves (Fig 1b). These temperature-limited growth curves reflect community-aggregate temperature dependencies consistently with the fact that our model captures acclimation at the community level through mean changes in the plastic trait $U$.

**Table 2. Ranges of temperature increase ($\Delta T_{obs}$) and corresponding ranges of symbiont abundance reduction ($\Delta S_{obs}$) derived from observational data of temperature-induced bleaching (see S1 Appendix, for additional details).**

| Region | $\Delta T_{obs}$ (°C) | $\Delta S_{obs}$ (%) |
|--------|-----------------------|----------------------|
| GBR    | 2–4                   | 10–95                |
|        | $\geq 4$              | 60–95                |
| SEA    | $\geq 1$              | 25–95                |
| CAR    | 1–3                   | 15–95                |
|        | 3–6                   | 35–95                |
|        | $\geq 6$              | 80–95                |

Eq 3 describes the temporal dynamics of the trait *U*. This trait is a physiological trait reflecting the energy that corals invest in the symbiotic relationship. By assuming that the temporal change of *U* is proportional to the fitness gradient of the coral, we capture the general feature of any adaptive process in which the trait of an organism moves towards values that increases fitness [59]. This is consistent with the "adaptive plasticity hypothesis" [60], which states that phenotypic plasticity maximises the fitness of a population in a variable environment (although in nature plastic responses do not always increase fitness and can actually be maladaptive). The constant of proportionality *N* (Eq 3) represents the speed with which corals move towards an optimal trait value, i.e. the one that maximises their fitness (Eq 4). Coral fitness, in our model, depends on many factors, including temperature (via coral growth, Eq 7). Thus corals acclimate to maximise energy gains under changing temperature conditions.

## Speed of acclimation

We estimated the speed of acclimation, for each studied region for the period 1970–2007 (Fig 2), by comparing the simulated relative coral abundance (i.e. percentage of coral abundance with respect to regional carrying capacity) with observed relative coral abundance (i.e. the mean percentage of coral cover with respect to the potential reef habitat estimated visually by the person who collected the data). For the observational data, we used previously collected data of changes in coral cover over time [61], see S2 Appendix, and integrated this dataset with observations from the GBR [62]. Observations of percentage coral cover are usually collected without any specification of the size of the potential reef habitat from which it was estimated and therefore might not reflect the regional coral carrying capacity. However, given that a potential reef habitat represents a snapshot of the total area that could be covered by corals, we assumed that the mean of all observations of coral cover reflects, qualitatively, a measure of coral abundance in relation to a carrying capacity (i.e. in relation to the total amount of coral cover that the considered region can sustain), see section Coral dynamics. The estimates included the following steps: (1) the model parameters were fixed to reported literature values (see Table 1), (2) the model was run with historical environmental temperature forcing (i.e. the WOD13 data only from 1970 to 2010, see S3 Appendix), and (3) the speed of acclimation

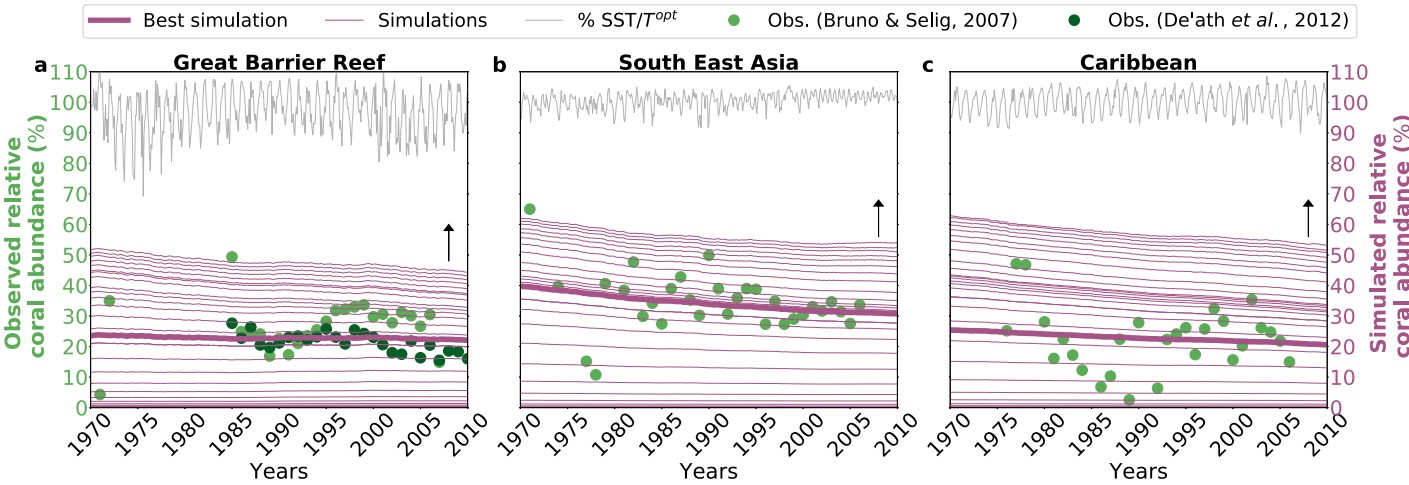

**Fig 2. Model simulations at different speeds of acclimation.** Model simulations (purple lines) are qualitatively compared to observations [61, 62], expressed as yearly median of coral abundances (green dots). The selected speed of acclimation, in each region (panels **a**, **b**, and **c**, respectively), is the one that produces results (thick purple line) consistent with observations. The arrows indicate the direction of increase in speed of acclimation and the thin grey lines indicate, for each region, environmental temperatures relative to corresponding optimal growth temperatures $T^{opt}$.

was adjusted to match model results with observations (Fig 2). The model results that compared best with observations were selected visually.

## Study regions and temperature forcing

We apply our model to three distinct coral-reef locations (Fig 1c): the Great Barrier Reef (GBR), South East Asia (SEA), and the Caribbean (CAR). For the Great Barrier Reef, we considered the region between 145° East and 165° East and between 10° South and 28° South. For South East Asia, we considered the region between 100° East and 137° East, and between 13° North and 10° South. The Caribbean region is between 65° West and 80° West and between 10° North and 20° North.

We collected Sea Surface Temperature (SST) data from the World Ocean Database 2013 (WOD13, https://www.nodc.noaa.gov/OC5/WOD13/) and considered only the data flagged as acceptable in quality by the WOD13 curators. For the future temperature scenarios, we considered the Representative Concentration Pathways (RCP) designed for the Coupled Model Intercomparison Project Phase 5 (CMIP5) [63]. Thus, we collected the temperature data of the RCP scenarios generated by the Max Planck Institute Earth System Model (MPI–ESM), compiled and maintained by the German Climate Computing Centre (DKRZ https://esgf-data.dkrz.de/search/cmip5-dkrz/). We considered the low RCP 2.6, the moderate RCP 4.5 and the high RCP 8.5 $CO_2$ emission scenarios at medium-resolution (MPI-ESM-MR). These scenarios predict that, relative to 1986–2005, temperatures can rise for the period 2081–2100 between 0.3 and 1.7°C under RCP 2.6; between 1.1 and 2.6°C under RCP 4.5, and between 2.6 and 4.8°C under RCP 8.5 [64]. The combined environmental temperature datasets (the WOD13 historical data from 1955 to 2010 and the future RCP scenarios from 2010 to 2100) used to force the model are presented in S3 Appendix.

In order to assess the effects of the long-term trend of global warming, we conducted model simulations under the hypothetical absence of short-term (monthly) temperature fluctuations, i.e. without bleaching. By visually comparing the model results obtained with bleaching and without bleaching, one can also evaluate, albeit qualitatively, the impact of bleaching events. The experiments without bleaching were performed by forcing the model with the annual averages of environmental temperatures for the period 1955–2100. The results of these experiments are presented in S4 Appendix.

## Simulations and sensitivity analysis

The mathematical model is coded in Python. In the code, small terms are added to the variables in the denominators in order to prevent divisions by zero. The simulations are structured as follows. We first consider a spin-up phase of 2000 years during which we let the model dynamics evolve under fixed temperatures (equal to the average temperatures of the period 1955–2000 in the three regions, which are 25.90°C, 28.45°C, and 27.57°C for, respectively, the Great Barrier Reef, the South East Asia, and the Caribbean). This spin-up phase removes numerical artefacts typically occurring at the beginning of a simulation due to non-linearities in the model equations. After this phase, the model has reached an equilibrium and we then introduce the temperature forcing for the period 1955–2100 to produce the actual long-term model results.

The initial conditions were chosen as follows: $C(t = 0) = 0.75 \cdot K_C$ cm$^2$; $U(t = 0) = 5 \cdot 10^{-7}$ energy cm$^{-2}$ month$^{-1}$; $S(t = 0) = 10^{-3}$ cells. These initial conditions refer to the beginning of the spin-up phase, 2000 years prior 1955. Not knowing the abundances of corals and symbionts in the regions of interest at this initial time, we adopted a conservative approach by setting the initial coral abundance to 75% of the coral regional carrying capacities ($K_C$), which we

estimated from suitable coral reef habitats (see subsection Coral dynamics). The initial conditions for $U$ and $S$ were chosen based on technical constraints in order to ensure that the model results are at equilibrium throughout the simulations and to avoid numerical crashes. However, the functional relationships between corals and symbionts in our model (e.g., the amount of symbiont abundance per coral abundance, $K_{smax}$, used to calculate the symbiont carrying capacity, see subsection Symbiont dynamics) are based on observational data. Therefore, despite the caveats, we are confident that, during the spin-up phase, the model results evolve towards realistic abundance levels for both symbionts and corals.

Given the uncertainties involved in modelling studies like this one, we conducted an in-depth analysis to explore the sensitivity of the model results to specific assumptions and to variations in the speed of acclimation and the other parameters. With respect to specific assumptions, we tested the model results in the absence of bleaching, as explained at the end of subsection Study regions and temperature forcing, and in the absence of acclimation (i.e. with $N = 0$ in Eq 3, which implies for corals a constant investment of energy into the symbiotic relationship). The results of these sensitivity analyses are presented in the next section and in S5 Appendix.

## Results

### Speed of acclimation

It is currently unknown whether the rate of coral acclimation, i.e. the speed with which corals acclimate to changing environmental conditions, is fast enough to match, and possibly offset, the current and future rates of warming. We do not have any information on the speed of coral acclimation and the physiological and ecological mechanisms with which coral reef communities may be able to acclimate are only poorly understood. Therefore, we provide a first, qualitative estimate of the speed of coral acclimation $N$ using our model in combination with observations on coral cover. Although coral cover data are influenced by a blend of processes and factors (e.g. multiple co-occurring environmental disturbances, ecological competition, etc.), they provide a preliminary workbench and a comparable metric across different reefs for first qualitative estimates. The observations show an overall decreasing trend in relative coral abundance, for the period 1970–2010 (Fig 2, green dots). In all regions, the simulated relative coral abundance (Fig 2, thin purple lines) increases with increasing speed of coral acclimation (Fig 2, upward arrow). We estimated speeds of acclimation $N = 5.54 \cdot 10^{-13}$, $N = 2.65 \cdot 10^{-13}$ and $N = 2.375 \cdot 10^{-13}$ for, respectively, the Great Barrier Reef, the South East Asia and the Caribbean. These numbers are very small because, according to the units of the model, the speed of coral acclimation reflects the amount of energy that the equivalent of 1 cm$^2$ of coral cover (i.e., an extremely small portion of a coral colony) invests into the symbiotic relationship every month (see Table 1).

In our model, the speed of acclimation depends on several factors, including (1) the equilibrium of the abundance levels reached during the spin-up phase, (2) the temperature-limited growth formulation, and (3) the observed relative coral abundances. Thus, corals exhibit different speeds of acclimation in the three regions (Table 1). This reflects the different acclimation capacities that coral communities with varying species compositions can have in different regions. The estimated speeds of acclimation allowed us to produce predictions for the dynamics of coral trait and coral and algal abundances under increasing bleaching events resulting from different emission scenarios.

### Future projections

Our model indicates that the cumulative number of bleaching events increases with increasing emissions in all regions (Fig 3a–3c). However, the rate of increase in bleaching events is less

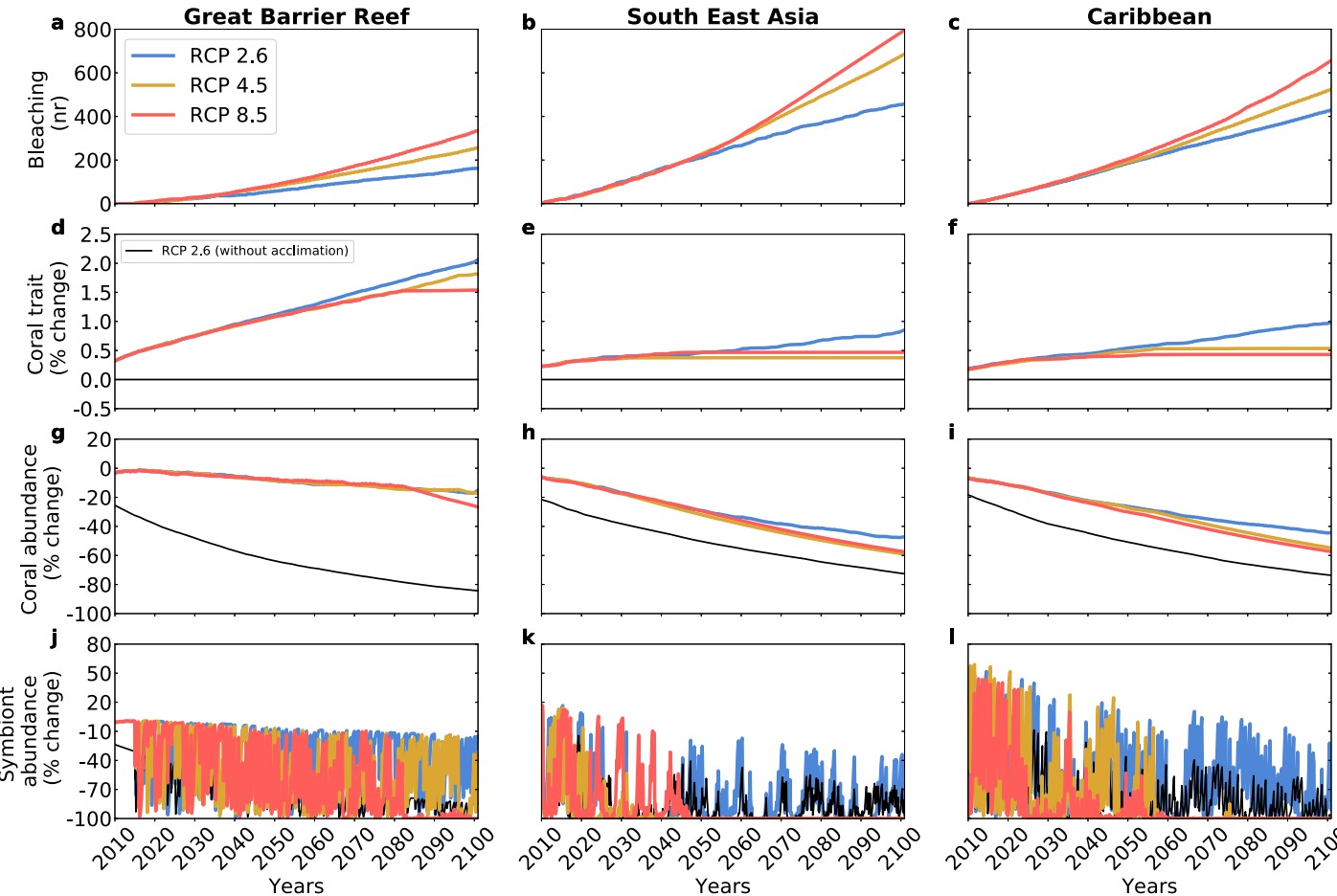

**Fig 3. Simulated trait and abundance dynamics in the three regions and under different RCPs. (a-c)** Cumulative number of bleaching assuming that these events occur whenever environmental temperature exceeds by 2˚C the optimal temperature for coral growth. **(d-f)** Coral energy investment trait. **(g-i)** Coral abundance and **(j-l)** symbiont abundance, relative to the period 1986–2005. Note that in South East Asia and in the Caribbean, the dynamics of coral trait and coral abundance under RCP 4.5 overlap with those under RCP 8.5. In the year 2010, the coloured lines and the black lines are at different levels because, in the runs with acclimation (coloured lines), corals are able to reach higher abundances during the spin-up phase and between 1955 and 2010, thus increasing the overall performance of the coral-algae complex.

pronounced in the Great Barrier Reef (Fig 3a) than in the other regions (Fig 3b and 3c). This allows corals of the Great Barrier Reef to better acclimate to the increasing temperature by increasing energy investments $U$ (Fig 3d–3f). The capacity of corals to acclimate, and thus to counteract bleaching, is lowest under scenario RCP 8.5 (Fig 3d–3f).

The increasing number of bleaching events is associated with declines in coral abundance in all regions, albeit the decline is least severe in the Great Barrier Reef (Fig 3g–3i). Bleaching events (Fig 3a–3c) are also characterised by fluctuations in symbiont abundance (Fig 3j–3l). Under RCP 4.5 and RCP 8.5, the symbiont populations of South East Asia and the Caribbean collapse by the year 2060 due to the high number of bleaching events in those regions. In our model, corals die at a fixed mortality rate of $0.83 \cdot 10^{-3}$ month$^{-1}$, implying that when symbionts are fully expelled corals will decline slowly. This is why neither fluctuations in symbiont abundance nor a full expulsion of symbionts produces a complete collapse of the coral communities (Fig 3g–3i).

Corals invest energy into the symbiotic relationship in order to "harvest" photosynthetic products from the symbionts. The benefits of investing energy into the symbiotic relationship

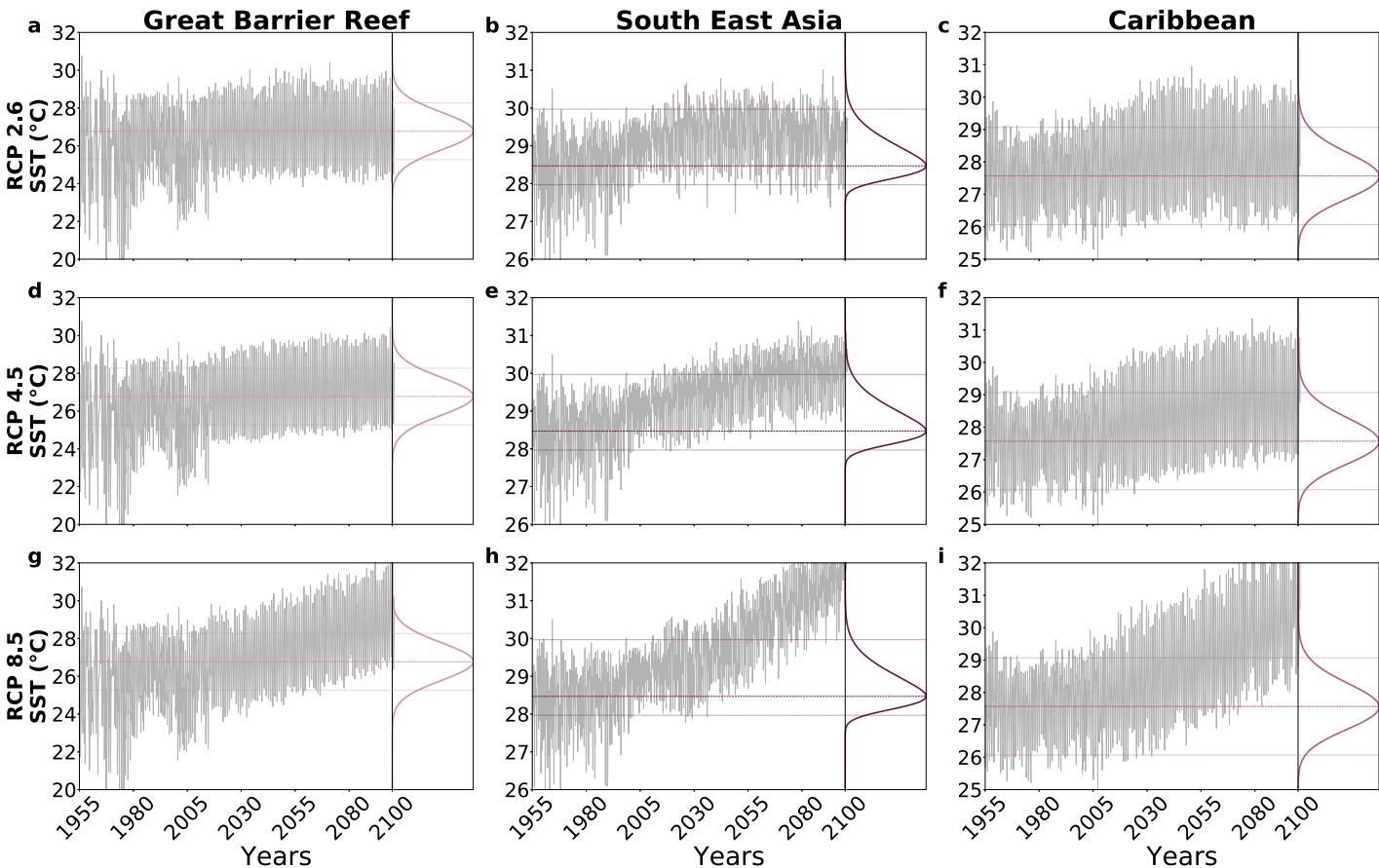

**Fig 4. Monthly temperature forcing scenarios and thermal-limited coral growth curves.** Low emissions (RCP 2.6) produce temperature trends (grey lines) that fall within the thermal-limited coral growth curves (coloured lines in the right side of each panel) in all regions (**a**-**c**). As emissions increase (RCP 4.5), the trends in environmental temperatures move away from the temperature optima $T^{opt}$ (marked by dashed lines), especially in the South East Asia and in the Caribbean (**d**-**f**) and even fall outside the thermal tolerance curves under RCP 8.5 (**g**-**i**). The dotted lines mark the limits of the coral thermal tolerances.

(represented by the coral gross growth, the positive term in Eq 4) manifest themselves in pluses following heat waves (see S6 Appendix). The benefits of the symbiotic relationship for corals change when the physiological responses that bolster the symbionts are activated by the corals during heat waves. However, benefits and costs are nearly anti-reciprocal (because of the nature of our modelling approach, according to which the trait evolves towards values that optimise fitness) and thus cancel each other out, producing a continuous, smooth dynamics in the coral trait (Fig 3d–3f).

The model runs conducted without acclimation confirms that the capacity of corals to respond to increasing temperatures is generally low (Fig 3d–3i). The acclimation capacity of corals is among the lowest under RCP 8.5 because, under this scenario, environmental temperature $T$ moves away from the thermal optima $T^{opt}$ and even reach values that are outside the temperature-limited coral growth curves (Fig 4). The acclimation capacity of corals is higher in the Great Barrier Reef, as compared to other regions, because in the Great Barrier Reef the temperature trend remains within the temperature-limited coral growth curves, at least under RCP 2.6 and RCP 4.5 (Fig 4).

Overall, for the period 2081–2100 relative to 1986–2005, our model predicts percentage declines in coral abundances by 15%, 12%, and 18% under, respectively, RCP 2.6, RCP 4.5, and RCP 8.5 in the Great Barrier Reef; by 47%, 55%, and 53% under, respectively, RCP 2.6,

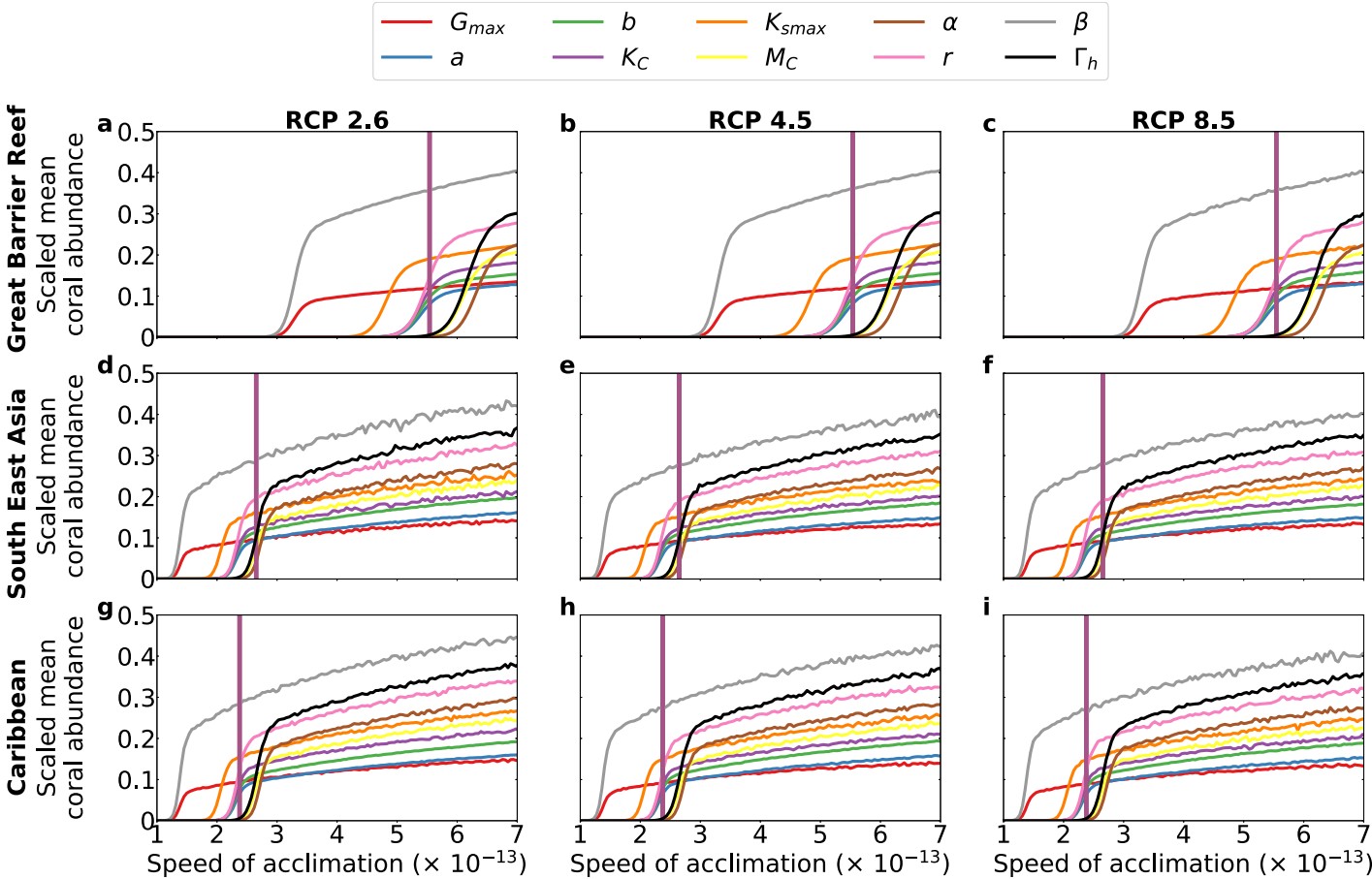

**Fig 5. Sensitivity to speed of acclimation *N* for +25% change in each parameter.** Simulations with bleaching. Vertical lines mark the speed of acclimation we estimated from coral cover data (see S2 Appendix).

RCP 4.5, and RCP 8.5 in South East Asia; and by more than 42%, 49%, and 52% under, respectively, RCP 2.6, RCP 4.5, and RCP 8.5 in the Caribbean.

## Model sensitivity to speed of acclimation

We performed simulations over a range of speeds of acclimation with and without bleaching. We scaled the resulting coral mean abundances in order to highlight the effects produced by ±25% change in the value of each parameter at a corresponding speed of acclimation. In all regions, coral abundances increase with increasing speeds of acclimation, regardless of the percent change in any of the model parameter considered and independently of the presence of bleaching (see Figs 5 and 6, for the case with bleaching, and S5 Appendix, for the case without bleaching).

## Discussion

The potential for coral communities to acclimate to global warming is critically important for the future of reef ecosystems. Yet, and despite major scientific efforts devoted to this aspect, we still know surprising little about it. The rate with which corals can acclimate to increasing temperatures is also a subject of controversy [65], perhaps unsurprisingly given the many different physiological and ecological mechanisms potentially involved (see [26, 28, 33]). Using our

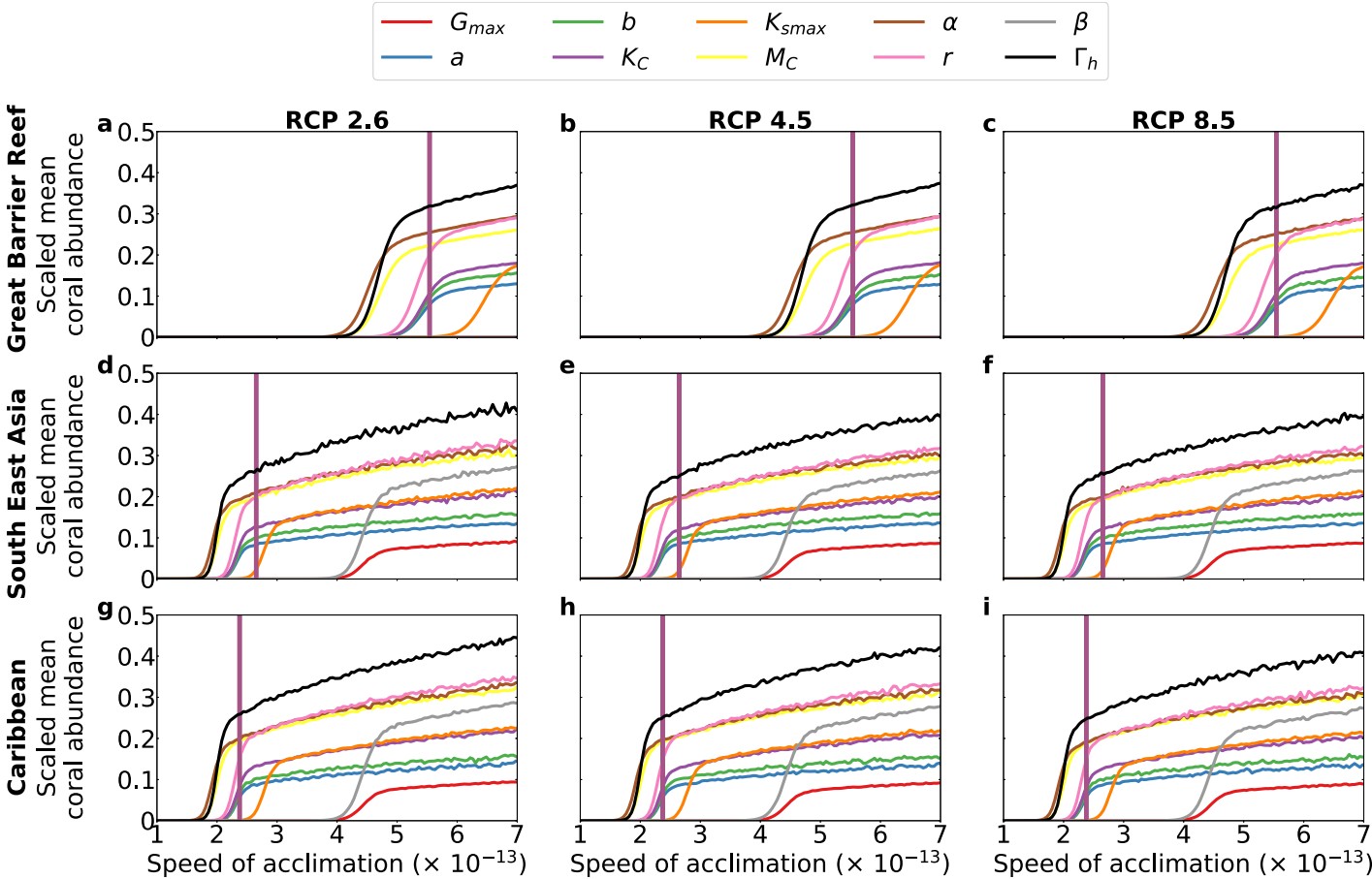

**Fig 6. Sensitivity to speed of acclimation *N* for −25% change in each parameter.** Simulations with bleaching. Vertical lines mark the speed of acclimation we estimated from coral cover data (see S2 Appendix).

model in combination with coral cover data [61, 62], we produce first-order estimates of the speeds of coral acclimation (*N*) for the three different regions considered in this study. Physiological acclimation in nature involves phenotypic variations that could be brought about by plastic changes in the physiology of the different populations composing the coral community or by shifts in species compositions. Our model does not resolve single species but it captures the effects that such shifts may have at the community level through changes in the average trait.

Being based on qualitative comparisons between model results and coral cover data and since most of coral cover data are not well resolved in terms of spatial and temporal coverages, these estimates remain somewhat crude. However, to our knowledge, the coral cover data we used constitute the best time-series currently available for attempting such preliminary estimates.

With our estimated speeds of acclimation and for the period 2081–2100 relative to 1986–2005, the model predicts up to 2% increase in energy investment *U* (e.g. in the Great Barrier Reef under RCP 2.6, Fig 3d) and up to 55% decrease in coral abundance (e.g. in South East Asia, under RCP 8. 5, Fig 3h). We should point out that the decreases in coral abundances shown by our results are rather conservative predictions because, for example, our model does not include coral starvation mechanisms under heat stress events [66]. Nonetheless, our results

indicate that the current rate of coral acclimation may not be sufficient to preserve coral reefs in the future, unless rapid genetic changes allow corals to shift their temperature-limited growth to a higher temperature optimum (see Fig 4 showing projected temperatures moving outside current temperature-limited coral growth curves under increasing emissions). Transplant experiments suggest that both genetic adaptation and acclimation can operate in some populations of fast-growing corals [26].

We also found that, under the hypothetical absence of short-term (monthly) temperature fluctuations, a substantial decline in coral abundance will occur in South East Asia and in the Caribbean (e.g. 27% and 37% decrease in coral abundance under, respectively, moderate RCP 4.5 and high RCP 8.5 emission scenarios, see S4 Appendix). Previous model projections [30, 31] showed that even if coral reefs had high thermal acclimation or adaptation capacities, they would still undergo long-term degradation. Consistently, our model indicates that, even if the conditions causing bleaching were to be mitigated, coral abundance would still decline due to the long-term trend component of global warming (see S4 Appendix). Although recent observations showed that coral reefs have already acclimated to an increase in temperature of 0.5˚C [67], the physiological mechanisms behind this change are not clear. A possibility is that changes in symbiont composition (towards more thermally tolerant symbionts) could have increased the thermal tolerance of the coral-algae complex [52] and thereby their thermal bleaching threshold.

Our sensitivity analysis shows that higher speed of acclimation $N$ would lead to higher coral abundance (Figs 5 and 6, and S5 Appendix). This indicates that our acclimation formulation is robust with respect to specific choices of parameter values because coral abundance always increases with increasing speed of acclimation, as expected, given that faster acclimation should lead to higher abundance yield. The model results are sensitive to parameters such as maximum coral growth rate $G_{max}$ (see S5 Appendix). This suggests that local management policies aiming at reducing nutrient runoff and pollution could be an effective strategy for mitigating anthropogenic impacts on coral growth [68] and, thus, for buffering the decline of coral communities. Additionally, the parameter $G_{max}$, corresponding to overall coral community growth, depends on the community composition because different coral species display different growth rates [42]. Shifts in species compositions (not captured by our model) might contribute to increase the value of $G_{max}$ and, consequently, mitigate the decline of coral-algae abundances. The endeavour of capturing inter-specific responses of corals to climate change is relevant but it would require a different modelling approach than that used here, for example, Agent-Based Modelling [65].

Our modelling approach is based on a generalised theory of acclimation and thus we conceptualised the coral trait as the energy that corals invest in the symbiotic relationship. This approach allows us to derive only a qualitative understanding of what might occur to corals under global warming. Laboratory data would be needed to equip the model with a less abstract formulation of the energy investment trait for example based on carbon and nitrogen fluxes. Alternatively, our model could be further developed to include a more detailed treatment of the physiological and biogeochemical processes involved in energy investment by following, for example, the principles of Dynamic Energy Budget theory [69, 70]. Additionally, our modelling approach considers coral and algal communities as single entities, despite evidence shows that the thermal acclimation of symbionts could be species-specific and that harbouring mixed algal populations could constitute an ecological advantage for corals [52]. The effects of different combinations of symbiont species and their shuffling under changing environmental temperature are thus promising avenues for future research.

Temperature increase is not the only problem faced by corals. Many anthropogenic stressors, such as eutrophication, ocean acidification and deoxygenation, concur to reduce coral

abundance over time [16, 71, 72]. In this respect, our results are quite conservative given our focus on temperature-related disturbances. However, we hope that our modelling study can foster research on the rates of coral acclimation, which is a key natural determinant of coral survival under global warming. Gobal solutions for reducing emissions, management policies for minimising local anthropogenic threats, and human-assisted evolution remain the ultimate strategies for protecting coral reefs because, according to our model results, their natural acclimation capacity alone will not be sufficient to offset the effects of global warming.

## Supporting information

**S1 Appendix. Levels of bleaching.**
(PDF)

**S2 Appendix. Speed of acclimation.**
(PDF)

**S3 Appendix. Forcing data.**
(PDF)

**S4 Appendix. Simulations without bleaching.**
(PDF)

**S5 Appendix. Sensitivity analysis.**
(PDF)

**S6 Appendix. Symbiotic relationship: Benefits versus costs for corals.**
(PDF)

## Acknowledgments

We are very grateful to John Bruno for providing us the dataset for global coral reef cover.

## Author Contributions

**Conceptualization:** Nomenjanahary Alexia Raharinirina, Agostino Merico.

**Data curation:** Nomenjanahary Alexia Raharinirina.

**Formal analysis:** Nomenjanahary Alexia Raharinirina, Esteban Acevedo-Trejos, Agostino Merico.

**Funding acquisition:** Nomenjanahary Alexia Raharinirina, Agostino Merico.

**Methodology:** Nomenjanahary Alexia Raharinirina, Esteban Acevedo-Trejos, Agostino Merico.

**Project administration:** Agostino Merico.

**Supervision:** Esteban Acevedo-Trejos, Agostino Merico.

**Writing – original draft:** Nomenjanahary Alexia Raharinirina, Agostino Merico.

**Writing – review & editing:** Nomenjanahary Alexia Raharinirina, Esteban Acevedo-Trejos, Agostino Merico.

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
