## [Decision Letter · Decision Letter 0]

14 Nov 2021

Dear Prof Merico,

Thank you very much for submitting your manuscript "Modelling the acclimation capacity of coral reefs to a warming ocean" for consideration at PLOS Computational Biology.

As with all papers reviewed by the journal, your manuscript was reviewed by members of the editorial board and by several independent reviewers. In light of the reviews (below this email), we would like to invite the resubmission of a significantly-revised version that takes into account the reviewers' comments.

Thank you for submitting your work to PLOS Computational Biology, overall, the paper is interesting and provides fresh insights on modeling for coral reef populations an important and understudied topic. The reviewers have provided ample feedback that is bound to make this manuscript better in the next revision. Key issues such as the assumptions of the model, distinction between acclimatization and genetic adaptation must be addressed. Please make the changes requested by the reviewers address them with a point-by-point response. In addition, the manuscript can benefit from a round of proof-reading to eliminate some of the lingering grammar mistakes and improve the overall readability. I will be looking forward to reading the revised manuscript.

Best,

Bishoy

We cannot make any decision about publication until we have seen the revised manuscript and your response to the reviewers' comments. Your revised manuscript is also likely to be sent to reviewers for further evaluation.

Sincerely,

Bishoy Kamel

Guest Editor

PLOS Computational Biology

James O'Dwyer

Deputy Editor

PLOS Computational Biology

Thank you for submitting your work to PLOS Computational Biology, overall, the paper is interesting and provides fresh insights on modeling for coral reef populations an important and understudied topic. The reviewers have provided ample feedback that is bound to make this manuscript better in the next revision. Key issues such as the assumptions of the model, distinction between acclimatization and genetic adaptation must be addressed. Please make the changes requested by the reviewers address them with a point-by-point response. In addition, the manuscript can benefit from a round of proof-reading to eliminate some of the lingering grammar mistakes and improve the overall readability. I will be looking forward to reading the revised manuscript.

Best,

Bishoy

Reviewer's Responses to Questions

**Comments to the Authors:**

Reviewer #1: In this paper, the authors develop a computational model of coral-symbiont dynamics in the context of rising sea surface temperatures to estimate the extent to which phenotypic plasticity can keep pace with global warming projections in three distinct coral reef systems: the GBR, SE Asia, and the Caribbean. The authors first formulated a general dynamical population model and then visually fitted predictions of this model to multi-decadal time series of coral cover in each of the three focal reef systems. This fitting provided system-specific estimates for rates of adaptation in a coral trait that translates to energetic investment in symbiotic zooxanthellae, estimates then used in future projections of long-term coral cover in each system under each of three region-specific future climatic warming projections. The authors find that the ability of coral communities to acclimate to future warming is generally limited but varies by region, with the GBR showing the greatest and the Caribbean the least capacity to adapt to keep pace with climatic projections. This work contributes a novel, empirically rooted computational modeling framework that provides important initial insights on the capacity for reef forming corals to use phenotypic plasticity to counter warming - a poorly understood phenomenon that is fundamental to our understanding of the ecology and conservation of coral reefs globally in the face of unprecedented anthropogenic threats.

I really enjoyed reading this paper. Aside from very minor grammatical issues throughout, it was easy to follow and was well motivated, methods and results clearly presented, and conclusions well justified by the findings. I also think that this type of global scale, data-motivated projection modeling is exactly the kind of work that is needed to set the stage to establish realistic, empirically rooted estimates for what the future will hold for critically threatened ecosystems. I commend the authors on the novelty and practicality of the work. I believe that a revised version of this work could be a highly cited addition to this journal. To this end, below I offer some comments and suggestions (with line number, where applicable) that I hope held improve the final version of the ms.

First, since coral cover is the only metric used in the time series that the model is being fitted to, I find myself wondering how much a signal of acclimation could be falsely attributed to a shift in coral species assemblage. For example, if temperatures increase over, say, 5 years in one of the study regions, and coral cover initially dips but then rebounds, this may fit the results of a model with acclimation and no species turnover when it is actually species turnover in this case that is driving the pattern. I appreciate that the authors qualify their findings in several instances by acknowledging both that this work is essentially a ‘first pass’ at estimating a very difficult, but crucial (from a conservation perspective) empirical metric (acclimation rate) and that other, not-modeled factors could be playing a significant role in the time series to which the model was fitted. But I think the authors could go a bit deeper here (e.g., discussing leading alternative drivers for the patterns observed and the likelihood that these would quantitatively or qualitatively affect the paper’s conclusions), to better contextualize their findings and the extent to which it seems likely that the model’s estimates for acclimation rate would be anywhere near reality. This additional content would also help paint a clearer picture for future work, by helping to prioritize the most timely extensions of this modeling framework.

Second, I think it would be helpful to have a bit more motivation for the mechanism of acclimation - how realistic is it for trait-dependent increases in coral energetic investment in symbionts to prevent bleaching? And what occurs physiologically to allow this to happen? A deeper foray into why this approach is realistic would help, I think, give credence to the overall paper results, which stem from these fundamental assumptions.

More specific / minor commoners:

Fig. 2 shows data juxtaposed to simulations for different acclimation rates, but it would be helpful to contextualize these plots a bit more by also showing SST over this same time period, perhaps absolute temps with a reference to each study region’s optimum value from Fig. 1b.

171: “We implement bleaching as a reduction in symbiont abundance when environmental temperature exceeds Topt, i.e. the optimal temperature for coral growth.” Table 2 then mentions that once the threshold temperature is exceeded, bleaching is determined by a draw from a uniform distribution. But isn’t bleaching severity correlated with temperature? Thus, I’m wondering: why not make bleaching a monotonic function of temperature, after the threshold is crossed, or, to maintain the stochastic element, make it a draw from a skewed distribution that reflects the bias of greater bleaching with greater temperatures? This seems reasonable, and so could be worth using, even as an alternative formulation to test the robustness of findings.

85: “…means that we always assume that the reef habitat has a maximal capacity of hosting an equal proportion of massive and branching corals.” Could you please unpack this statement a bit more? It’s not clear to me how the conversion factors mentioned in the preceding sentence ensure equal proportions of the two coral growth morphologies. I also wonder whether this assumption of equal proportions is also one that could be relaxed (perhaps based on region-specific relative species abundance data) to test the robustness of findings.

110: missing ‘cm’ after 10 and 0.83

152: “…host to control the nutrient flux. The latter ensures that the symbiont population never exceeds the hosting capacity of the corals.”

The authors later mention eutrophication as a potentially important additional factor to consider in the Discussion, but it could be useful to point out here how this assumption can be violated: data suggest that eutrophication can elicit a breakdown in the coral-zooxanthellae mutualism, whereby uncontrolled symbiont growth can actually depress coral growth (e.g., Gil, MA. “Unity through nonlinearity: A unimodal coral-nutrient interaction”. Ecology 2013). It’s outside the scope of this paper to speculate too much on potential indirect environmental effects that could interact with coral acclimation to rising temperatures, but it is also interesting to consider that eutrophication is often driven by runoff events that can increase in magnitude and frequency with warming, but in a region-specific manner - perhaps a cool future extension of this model.

201: ‘a shifts’ should be ‘an increase’

221: “However, given that a potential reef habitat represents a snapshot of the total area that could be covered by corals, we assumed that the mean of all observations of coral cover reflects, qualitatively, a measure of coral abundance in relation to a carrying capacity (i.e. in relation to the total amount of coral cover that the considered region can sustain).” From this statement, it’s not clear to me how the region-specific coral carrying capacity was determined. Is the carrying capacity the mean of all observations of coral cover? This would seem biased in favor of a lower carrying capacity than reality, given the high disturbance frequencies and magnitudes of these systems. Please provide further detail.

261: “…we conducted model simulations without bleaching.”

It would be helpful to describe the motivation for the modeling without bleaching.

266: “After this phase, we introduced the temperature forcing described in the previous section (for the period 1955–2100) in order to produce…”

Should 1955 be 2000, since this would be after the previously mentioned phase of 1955-2000?

Reviewer #2: This paper presents a model to explore a crucial topic in coral reef population ecology: the capacity for acclimatization in response to climate change. I was particularly impressed by the careful approach to model parameterization while maintaining generality.

My most substantive concern is that, to truly quantify the role of acclimatization, the paper needs to compare the results Fig. 3 to the case without any acclimatization (N=0). Without this comparison to isolate the effect of acclimatization, the authors cannot make conclusions about the specific role of acclimatization in their outcomes (e.g., lines 307-308), which is the primary goal of this paper. This should be a relatively straightforward comparison to do with the existing model.

For the model structure, it took me a while to understand why the benefit of the plastic trait U, the energy investment in symbiosis, appears in the coral growth function (in dC/dt) rather than the symbiont dynamics (dS/dt). If one interprets this trait as physiological responses that bolster the symbionts during marine heat waves (e.g., heat shock protein up-regulation; as described on lines 198-201), then one would expect the benefit of greater U to come in through lower symbiont mortality during marine heat waves. I think the reason why the authors chose the approach they did is because they needed both the costs and benefits of U to be in the same function in order to have one function for taking the derivative to calculate the selection differential that determines the trait change dynamics (dU/dt); the costs do belong in the coral dynamics due to energy investment effects, so then the authors put in the benefit in terms of coral benefits from symbionts. Which I can understand from a tractability standpoint, but irregardless of whether this or another explanation is the right one, the authors need to clarify why they're taking the approach they do when they first develop the model (in the Coral dynamics section of the Methods), and discuss the potential effects of this assumption (e.g., the effect of benefits occurring continuously rather than in pulses during marine heat waves, and what types of acclimatization that does and doesn't represent) in the Discussion (somewhere around lines 402-405 where they're discussing phenomenological vs. mechanistic modeling approaches might make sense). If at all possible, it'd also be great to see a functional sensitivity analysis comparison to a model structure where the benefits accrue in the symbionts rather than the coral, but I understand that this might not be mathematically tractable.

For the model analyses, in addition to adding an analysis without acclimatization for comparison, the paper would be stronger if the authors had the sensitivity analysis (currently in the appendix, Figs. S6-S9) in the main text, as it recognizes uncertainty in the parameterization and leads to some of the more interesting conclusions about what most influences acclimatization potential. This doesn't have to be all four sensitivity analysis figures; the ones with bleaching are more relevant than the ones without, and the results were pretty consistent across climate scenarios so the authors could chose one climate scenario to illustrate the sensitivity analysis in the main text and put the remaining RCPs in the appendix. This would be even better, and the results presented more simply, if the authors took a more formal approach to sensitivity analysis, whether as a local sensitivity analysis or global sensitivity analysis as described in:

Cariboni, J., Gatelli, D., Liska, R., & Saltelli, A. (2007). The role of sensitivity analysis in ecological modelling. Ecological modelling, 203(1-2), 167-182.

plus see:

Harper, Elizabeth B., John C. Stella, and Alexander K. Fremier. "Global sensitivity analysis for complex ecological models: a case study of riparian cottonwood population dynamics." Ecological Applications 21.4 (2011): 1225-1240.

for some different ways to visualize global sensitivity analysis outputs.

Three points on the motivation for this modeling exercise in the Introduction:

1. While the Abstract and Author summary both start with establishing the threat that climate change poses to tropical coral reefs before getting into responses to climate change via acclimatization and adaptation, the Introduction launches straight into those responses. Before this, at the onset of the Introduction, the authors should have a short paragraph that sets up the impact climate change has on corals (with relevant references) before getting to acclimatization and adaptation potential.

2. For the motivation of focusing on acclimatization vs. genetic adaptation, I disagree with the authors' claim that that the long generation times of corals make evolution less relevant (lines 12-14, and repeated elsewhere: 3rd sentence of the Abstract, lines 67-69, 193-195, and 370-371), as evidence has now been mounting for decades that evolution can occur on ecological time scales (this body of research even now has its own field, eco-evo dynamics; see, e.g. Schoener 2011, Science 331:426-429). Even though corals are long-lived as the authors say, a selective sweep takes only one extreme event, such as a bleaching event, for gene frequencies to shift, and the symbionts within corals, which can also evolve, have very high turnover rates. Really, acclimatization vs. genetic genetic adaptation is a false dichotomy: rather than either/or, both have the potential to play a role in future coral dynamics under climate change, and there's no need to claim one as unimportant to say the other is important. Instead, the authors could just make the point that both acclimatization and genetic adaptation might play a role in future coral persistence, and while a few models have quantified evolutionary capacity, quantifying acclimatization capacity as well can help further understand overall coral adaptive potential. Note that, once the authors have established the motivation for why one might look to quantifying acclimatization capacity in the Introduction, there's no need to repeat the motivation elsewhere in the manuscript (e.g., lines 66-69, 193-195).

3. In both the Introduction and Discussion, I think there's a missed opportunity here to connect to the idea of "pre-exposure" (also called pre-conditioning, stress-hardening, and induced acclimatization, among other names) that is gaining interest in the field of coral reef restoration as a way to promote the likelihood of outplanted corals persisting under future climate change. See Chapter 3 on Physiological Interventions of the 2018 National Academies report on "A Research Review of Interventions to Increase the Persistence and Resilience of Coral Reefs", available at:

https://www.nationalacademies.org/our-work/interventions-to-increase-the-resilience-of-coral-reefs

for a review of this approach. Quantification of acclimatization potential, as is done in this paper, can help inform the capacity for this management intervention to affect coral persistence under future climate change. In addition, might the model developed here provide a framework for the types of decision-support tools described in the second report linked above (on "A Decision Framework for Interventions to Increase the Persistence and Resilience of Coral Reefs"), which can help managers navigate whether and how to implement interventions like pre-exposure?

The model could use greater clarity in presentation in a few places. In particular, somewhere in the Bleaching subsection of the Methods, it'd be helpful to have a mathematical expression for the implementation of bleaching with relevant parameters defined. Are the authors essentially having discrete bleaching events within the continuous-time model, such that this is a pulse-impulsive or semi-discrete model as described in:

Mailleret, L., & Lemesle, V. (2009). A note on semi-discrete modelling in the life sciences. Philosophical Transactions of the Royal Society A: Mathematical, Physical and Engineering Sciences, 367(1908), 4779-4799

? In addition, equations (1)-(6) would be clearer if all state variable dependencies were consistently specified for all functions, e.g. F(U,S,C) instead of F(I, kappa*E,C) in eq. 1, K_S(C) instead of K_S in eq. 2, and kappa(S)E(U) instead of kappa*E in eq. 4 (with these state-variable-dependency specifications elsewhere in the text wherever relevant), perhaps with temperature T dependency specified where relevant as well.

Additional comments:

First sentence of the Introduction on line 1: in addition to adaptation and acclimatization, there is a third commonly-invoked response that organisms might have to climate change, range shifts, which tend to get discussed less in the context of coral reefs but are one of the most frequently-observed responses to climate change across systems:

Parmesan, C. (2006). Ecological and evolutionary responses to recent climate change. Annu. Rev. Ecol. Evol. Syst., 37, 637-669.

Line 60: "adaptive dynamics" invokes a specific invasion analysis approach to modeling evolutionary change, please use a different phrase here.

Line 65: "plastic trait" would be a more precise label than "adaptive trait".

Lines 161-171 are repetitive to the Introduction, and spending this amount of space on DHMs in the Methods might mislead a reader into thinking that the authors are going to take a DHM approach. I would start with the approach taken here, then briefly justify it.

Line 209: this could be more clearly framed as an assumption that plasticity is adaptive in the case of this model, as in reality plasticity is not always adaptive, especially when human-driven environmental change leads to "evolutionary traps".

Section on "Simulations and sensitivity analysis": what are the initial conditions for the simulations?

Lines 275-285: this text is repetitive to the Introduction.

Line 333: typo in "important".

Line 349-350: this sentence is a fragment.

Lines 400-401: "theory of adaptation" should be "theory of acclimatization"; a generalized theory of adaptation would have both genetic and plastic responses.

In addition, this discussion of mechanistic vs. phenomenological approaches to modeling might benefit from the example of dynamic energy budget models as a mechanistic approach in contrast to the phenomenological approach here; note that DEB models have been applied to corals:

Cunning, R., Muller, E. B., Gates, R. D., & Nisbet, R. M. (2017). A dynamic bioenergetic model for coral-Symbiodinium symbioses and coral bleaching as an alternate stable state. Journal of Theoretical Biology, 431, 49-62.

Muller, E. B., Kooijman, S. A., Edmunds, P. J., Doyle, F. J., & Nisbet, R. M. (2009). Dynamic energy budgets in syntrophic symbiotic relationships between heterotrophic hosts and photoautotrophic symbionts. Journal of Theoretical Biology, 259(1), 44-57.

For the discussion of the sensitivity analysis, note that a potential implication of the high sensitivity to coral growth (lines 406-414) could be that local management approaches that might buffer coral growth, such as control of the runoff of pollutants that affect coral growth as reviewed in:

Fabricius, K. E. (2005). Effects of terrestrial runoff on the ecology of corals and coral reefs: review and synthesis. Marine pollution bulletin, 50(2), 125-146.

might then increase acclimatization potential to global climate change.

I appreciate that the authors discussed a few of the model assumptions throughout the Discussion. Another assumption to consider discussing is whether and how the model represents the potential for constraints to plasticity or acclimatization potential.

Reviewer #3: The authors model corals acclimating to increasing temperatures, where corals acclimate by changing the amount of energy they invest in the symbiosis. Energy investment can potentially make up for symbionts lost due to bleaching, by increasing the per-symbiont benefit of the symbiosis to the coral. The authors build a differential equation model that tracks the growth and energy investment of a single coral and symbiont. They investigate corals from three regions (the Great Barrier Reef, South East Asia, and the Caribbean) under three different temperature scenarios, and find that corals on the Great Barrier Reef are best able to use acclimation to survive, but none of the three regions' corals could survive via acclimation alone in the hottest warming scenario.

Coral acclimation is a topic of great interest for conservation, and the authors' focus on using incorporating real warming and trait data to understand the ability of coral acclimation to preserve reefs is likely to be of interest to many readers. The model and results are presented very clearly, and the authors do an excellent job of explaining the model's purpose and future uses.

There is a potential issue in the function for the temperature-dependent growth rate of the symbiont. The authors model the maximum symbiont growth rate as increasing exponentially with temperature. This relationship comes from studies across multiple species, where the maximum growth rate over all species at a particular temperature follows this pattern. However, individual species's growth rates probably do not follow this pattern (the Baskett et al. paper they cite models them as decreasing sharply after a certain temperature). For the model, this seems to imply that the coral is swapping out its symbionts every time the temperature changes, to always have a species that has the ability to achieve the maximum growth rate a particular temperature. I know the model is intended to provide information about populations, but since the differential equations so closely resemble the interaction between a single coral and symbiont, I think there are probably some consequences of modeling symbiont growth this way. In particular, the symbiont growth rate may be lower than is modeled (because most symbionts will not actually be growing at this rate). I think this decision and its consequences should be discussed.

Other comments:

I think it would be really helpful for gaining an intuitive understanding of the model to be able to see how the benefit of energy investment changes with temperature. A plot of the optimal investment for one of the warming scenarios would be nice, if that is possible to make.

The authors refer to the model as an adaptive dynamics model, which I think might be confusing to readers. To me adaptive dynamics is a technique for modeling evolution, and this model is important because it does not include evolution. Also, adaptive dynamics seems to me to require selection and some kind of inheritance mechanism. Possibly there is an argument that some sort of selection is present (maybe between polyps?). I think it would also be reasonable to say that this is a sensible acclimation rate to assume for other reasons. (It seems like a pretty sensible way to try to match your environment!)

It seems to me that by changing the strength of the symbiotic feedback (beta) and and the coral exponential cost parameter (r), it is possible to scale energy (and thus rate of acclimation) values. I don't think this is a problem for the model, but I do think this means it would be helpful to give some intuition for what an energy investment of a particular amount means to the model coral (maybe something like how much of its growth is it investing in the symbiont?), and why the beta and r values were chosen.

The temperature-dependent host growth rate is derived from coral distribution data. Intuitively, it makes sense that if no corals can grow at a temperature, their growth rate at that temperature is zero. I think a sentence or two explaining how to get from the coral distribution to the growth rate function in the nonzero growth rate case would be nice.

I just skimmed the code, but it looks like some small terms are being added to the denominators of some functions to prevent division by 0. If so, it would be good to mention that in the methods section.

**Have the authors made all data and (if applicable) computational code underlying the findings in their manuscript fully available?**

Reviewer #1: Yes

Reviewer #2: Yes

Reviewer #3: Yes

PLOS authors have the option to publish the peer review history of their article (what does this mean?). If published, this will include your full peer review and any attached files.

Reviewer #1: **Yes: **Mike Gil

Reviewer #2: No

Reviewer #3: No
---

## [Decision Letter · Decision Letter 1]

5 Apr 2022

Dear Prof Merico,

Thank you very much for submitting your manuscript "Modelling the acclimation capacity of coral reefs to a warming ocean" for consideration at PLOS Computational Biology. As with all papers reviewed by the journal, your manuscript was reviewed by members of the editorial board and by several independent reviewers. The reviewers appreciated the attention to an important topic. Based on the reviews, we are likely to accept this manuscript for publication, providing that you modify the manuscript according to the review recommendations.

Dear Dr. Merico and Co-authors,

Thanks for re-submitting the manuscript and addressing all the reviewer comments. Overall, I think the manuscript has improved and is shaping up to be an important contribution to the field. Before accepting it for final publication, there are few minor issues that still need to be addressed. Please see the attached feedback from the reviewers and provide a new modified draft with the requested edits.

As the reviewers mention there is some ambiguity regarding the use of “acclimation at the community level” and “plasticity” in some of the contexts in the manuscript, which warrants more clarification and discussion of these terms or their intended use in the text.

Thanks,

Bishoy

Sincerely,

Bishoy Kamel

Guest Editor

PLOS Computational Biology

James O'Dwyer

Deputy Editor

PLOS Computational Biology

[LINK]

Dear Dr. Merico and Co-authors,

Thanks for re-submitting the manuscript and addressing all the reviewer comments. Overall, I think the manuscript has improved and is shaping up to be an important contribution to the field. Before accepting it for final publication, there are few minor issues that still need to be addressed. Please see the attached feedback from the reviewers and provide a new modified draft with the requested edits.

As the reviewers mention there is some ambiguity regarding the use of “acclimation at the community level” and “plasticity” in some of the contexts in the manuscript, which warrants more clarification and discussion of these terms or their intended use in the text.

Thanks,

Bishoy

Reviewer's Responses to Questions

**Comments to the Authors:**

Reviewer #1: I appreciate the efforts the authors put into their revisions in response to my and the other reviewers’ comments. I believe the manuscript is clearly improved. The issue I have with this version is that I think the authors can and should be more up front about the fact that while they motivate this work around the need to understand ‘coral acclimation’, their model - as they acknowledge in the first paragraph of the discussion - deals with changes in phenotypes at the community level, something one could define as ‘acclimation at the community level’, I suppose. I struggle with this, because - again, as the authors acknowledge - species turnover is unavoidably a potential major mechanism underlying such ‘acclimation at the community level’, though this is not a mechanism one would think of as falling into the category of ‘acclimation’, since we (and the authors, in the Intro), tend to define acclimation at the organismal/species (not community) level. I think it would be better to be much more up front about this modeling study’s limitations with respect to assessing the capacity for ‘coral acclimation’ to respond to climate change. I’d recommend clarifying in the Abstract and Intro that this study focuses on the community level (and, thus, cannot isolate the effects of organismal adaptation), at which species turnover can (and, according to recent studies by Terry Hughes and colleagues working in the GBR, likely does) play a substantial role in shaping community phenotypic shifts in coral responding to environmental changes over time. For example, in the Abstract and Intro, the authors discuss the capacity for corals to acclimate via phenotypic plasticity, but to not confuse readers about the scale of inquiry for the actual study, I’d argue it would be more appreciate to discuss ‘coral community acclimation’, or some other term, and clarify that what we traditionally think of as acclimation or adaptation cannot be explicitly examined with the proposed model and that species turnover could be a critical driver.

Reviewer #2: The reviewers have mostly addressed my previous comments, especially with the added simulations without plasticity as a baseline for comparison, as well as the added figure S6 to illustrate the benefit and cost dynamics. However, some lingering issues remain from the new changes and unresolved previous comments (all line numbers refer to the revised manuscript without tracked changes):

1. While I agree with Reviewer #1's point about shifts in community composition, I disagree with the changes made in response: with the model setup, a shift in community composition would change parameters related to coral type, such as Topt and Gmax (as mentioned on line 452-453), perhaps with some changes in N too if different coral species had different plasticity levels, but I expect that this is swamped by the other differences. Therefore, instead of adding the claim that the model captures changes in community composition (lines 345 and 403-404), I support the original reviewer feedback of adding this as a caveat, i.e. recognizing that the acclimatization rates might be over-estimated because the model doesn't capture the effects community shifts on coral cover dynamics. I also support Reviewer #1's suggesting to include sensitivity to a couple of different values for the percent of each coral type present among the sensitivity analyses, which the authors had declined to do.

2. With moving some of the sensitivity analysis results to the main text in Figs. 5-6 as I had suggested, the authors also need to add Results text about these figures in the main text, which is currently missing.

3. While the authors indicate their agreement with my previous point about acclimation and genetic adaptation both playing a role in coral dynamics, and the false dichotomy inherent to presenting these as an either/or, the changes to the text do not reflect this: the framing language of "however", "alternately", and "although", as well as the continued mention of generation times (which, as I mentioned before, do not necessarily mean evolution is slow given the capacity for selective sweeps) and ecological vs. evolutionary time scales (which applies to macroevolutionary processes of speciation, not microevolutionary process of changes in gene frequency), on lines 29-31, 424-426, and in the abstract third sentence still imply that these are alternate, not co-occurring dynamics, and discounts the potential role of evolution. Please rephrase to recognize that both have the potential to occur, which can still motivate this study, as some studies have explored the role of genetic adaptation, but a mechanistic investigation into the role of acclimation is less well understood (with an eventual goal, hopefully, of including both to understanding their interaction, but understanding each in turn is a reasonable starting point and important to establishing their potential relative contributions before such integration). Also, in lines 36-38, note that the rolling-window approach to approximating adaptive and acclimation potential in these papers does not have an assumed-in bleaching threshold as implied here, but rather derives the threshold from the climatology, in particular the most recent mean of maximal monthly temperatures.

4. The added text about stress-hardening on lines 465-468, a point that I had raised in my previous review, is too vague to be useful. Please either (a) be specific about stress hardening (using that term and defining what it involves) as a potential management application, including citations on this approach, and how this model might inform that approach, or (b) delete this addition. For option (a), note that the addition is out of place: it is a sentence about management implications in the midst of a paragraph about model assumptions. Therefore, if included, it should be moved to a more appropriate place in the discussion.

5. For the data-based bleaching simulations: even if the parameters were estimated empirically, those parameters were somehow used in a mathematical expression (represented by code) to translate into their effect on coral dynamics. Please provide this information for full methods clarity.

6. To clarify my earlier comment about adaptive plasticity: what this text (lines 225-236) needs clarity on is the idea that plasticity is always perfectly adaptive, as represented in the model, is an assumption and does not always occur in reality, as, in reality, plasticity can sometimes decrease fitness, such as in the case of evolutionary traps. The current text could be misinterpreted to mean that the idea that plasticity is always adaptive is a biological truth.

7. For the addition about nutrient runoff and pollution effects on coral growth stemming from my previous feedback (lines 449), "controlling" doesn't make sense here; something like "buffering", "protecting", or (my preference) "mitigating anthropogenic impacts on" would work better (and why is this in quotes?).

8. On line 394, do you mean "acclimate" instead of "adapt"?

Reviewer #3: The authors addressed my comments, and there is just a small thing that I think should be added: I think in the methods the authors should say that the symbiont growth function comes from a curve of multiple species' thermal optima. The authors talk about modeling symbiont communities in the discussion, but I think it would be good to have this information about the function in the methods where the function is first described. Because the model partly has the purpose of being the "parent" of future models, it's likely that future readers will spend a lot of time with the methods section and would benefit from having all the information together.

I really liked the supplementary plots of the costs and benefits of the symbiosis. Figure S10 and all of section S6 helped me gain a better understanding of the model.

* line 49: "coral's" should possibly be "corals'"?

* line 386: "because of the acclimation dynamics nature of our modeling approach." I think removing "acclimation dynamics" from this phrase would make it clearer.

**Have the authors made all data and (if applicable) computational code underlying the findings in their manuscript fully available?**

Reviewer #1: Yes

Reviewer #2: Yes

Reviewer #3: Yes

PLOS authors have the option to publish the peer review history of their article (what does this mean?). If published, this will include your full peer review and any attached files.

Reviewer #1: No

Reviewer #2: No

Reviewer #3: No

Figure Files:

Data Requirements:

Reproducibility:

References:

---

## [Editor Report · Decision Letter 2]

12 Apr 2022

Dear Prof Merico,

We are pleased to inform you that your manuscript 'Modelling the acclimation capacity of coral reefs to a warming ocean' has been provisionally accepted for publication in PLOS Computational Biology.

Best regards,

Bishoy Kamel

Guest Editor

PLOS Computational Biology

James O'Dwyer

Deputy Editor

PLOS Computational Biology

Dear Dr. Merico and co-authors,

Thanks for providing a revised manuscript, all minor issues have been addressed and the manuscript is now ready for publication.

Congratulations on this important and timely body of work.

Regards,

Bishoy

---

## [Editor Report · Acceptance letter]

2 May 2022

PCOMPBIOL-D-21-01526R2 

Modelling the acclimation capacity of coral reefs to a warming ocean

Dear Dr Merico,

I am pleased to inform you that your manuscript has been formally accepted for publication in PLOS Computational Biology. Your manuscript is now with our production department and you will be notified of the publication date in due course.

With kind regards,

Agnes Pap
